# *Devign*: Effective Vulnerability Identification by Learning Comprehensive Program Semantics via Graph Neural Networks

Yaqin Zhou[1], Shangqing Liu[1, *], Jingkai Siow[1], Xiaoning Du[1, *], and Yang Liu[1]

[1]Nanyang Technological University
[1]{yqzhou, shangqin001, jingkai001, xiaoning.du, yangliu}@ntu.edu.sg
[*]Co-corresponding author

## Abstract

Vulnerability identification is crucial to protect the software systems from attacks for cyber security. It is especially important to localize the vulnerable functions among the source code to facilitate the fix. However, it is a challenging and tedious process, and also requires specialized security expertise. Inspired by the work on manually-defined patterns of vulnerabilities from various code representation graphs and the recent advance on graph neural networks, we propose *Devign*, a general graph neural network based model for graph-level classification through learning on a rich set of code semantic representations. It includes a novel *Conv* module to efficiently extract useful features in the learned rich node representations for graph-level classification. The model is trained over manually labeled datasets built on 4 diversified large-scale open-source C projects that incorporate high complexity and variety of real source code instead of synthesis code used in previous works. The results of the extensive evaluation on the datasets demonstrate that *Devign* outperforms the state of the arts significantly with an average of 10.51% higher accuracy and 8.68% F1 score, increases averagely 4.66% accuracy and 6.37% F1 by the *Conv* module.

## 1   Introduction

The number of software vulnerabilities has been increasing rapidly recently, either reported publicly through CVE (Common Vulnerabilities and Exposures) or discovered internally in proprietary code. In particular, the prevalence of open-source libraries not only accounts for the increment, but also propagates impact. These vulnerabilities, mostly caused by insecure code, can be exploited to attack software systems and cause substantial damages financially and socially.

Vulnerability identification is a crucial yet challenging problem in security. Besides the classic approaches such as static analysis [1, 2], dynamic analysis [3–8] and symbolic execution, a number of advances have been made in applying machine learning as a complementary approach. In these early methods [9–11], features or patterns hand-crafted by human experts are taken as inputs by machine learning algorithms to detect vulnerabilities. However, the root causes of vulnerabilities vary by types of weaknesses [12] and libraries, making it impractical to characterize all vulnerabilities in numerous libraries with the hand-crafted features.

To improve usability of the existing approaches and avoid the intense labor of human experts on feature extraction, recent works investigate the potential of deep neural networks on a more automated way of vulnerability identification [13–15]. However, all of these works have major limitations in learning comprehensive program semantics to characterize vulnerabilities of high diversity and

complexity in real source code. First, in terms of learning approaches, they either treat the source code as a flat sequence, which is similar to natural languages, or represent it with only partial information. However, source code is actually more structural and logical than natural languages and has heterogeneous aspects of representation such as Abstract Syntax Tree (AST), data flow, control flow and etc. Moreover, vulnerabilities are sometimes subtle flaws that require comprehensive investigation from multiple dimensions of semantics. Therefore, the drawbacks in the design of previous works limit their potentiality to cover various vulnerabilities. Second, in terms of training data, part of the data in [14] is labeled by static analyzers, which introduced high percentage of false positives that are not real vulnerabilities. Another part, like [13], are simple artificial code (even with "good" or "bad" inside the code to distinguish the vulnerable code and non-vulnerable code) that are far beyond the complexity of real code [16].

To this end, we propose a novel graph neural network based model with composite programming representation for factual vulnerability data. This allows us to encode a full set of classical programming code semantics to capture various vulnerability characteristics. A key innovation is a new *Conv* module which takes as input a graph's heterogeneous node features from gated recurrent units. The *Conv* module hierarchically chooses more coarse features via leveraging the traditional convolutional and dense layers for graph level classification. Moreover, to both testify the potential of the composite programming embedding for source code and the proposed graph neural network model for the challenging task of vulnerability identification, we compiled manually labeled data sets from 4 popular and diversified libraries in C programming language. We name this model *Devign* (Deep Vulnerability Identification via Graph Neural Networks).

- In the composite code representation, with ASTs as the backbone, we explicitly encode the program control and data dependency at different levels into a joint graph of heterogeneous edges with each type denoting the connection regarding to the corresponding representation. The comprehensive representation, not considered in previous works, facilitates to capture as extensive types and patterns of vulnerabilities as possible, and enables to learn better node representation through graph neural networks.
- We propose the gated graph neural network model with the *Conv* module for graph-level classification. The *Conv* module learns hierarchically from the node features to capture the higher level of representations for graph-level classification tasks.
- We implement *Devign*, and evaluate its effectiveness through *manually* labeled data sets (*cost around 600 man-hours*) collected from the 4 popular C libraries. We make two datasets public together with more details (https://sites.google.com/view/devign). The results show that *Devign* achieves an average 10.51% higher accuracy and 8.68% F1 score than baseline methods. Meanwhile, the *Conv* module brings an average 4.66% accuracy and 6.37% F1 gain. We compare *Devign* with well-known static analyzers, where *Devign* outperforms significantly with a 27.99% higher average F1 score for all the analyzers and on all the datasets. We apply *Devign* to 40 latest CVEs collected from the 4 projects and get 74.11% accuracy, manifesting its usability in discovering new vulnerabilities.

## 2 The *Devign* Model

Vulnerability patterns manually crafted with the code property graphs, integrating all syntax and dependency semantics, have been proved to be one of the most effective approaches [17] to detect software vulnerabilities. Inspired by this, we designed *Devign* to automate the above process on code property graphs to learn vulnerable patterns using graph neural networks [18]. The *Devign* architecture is shown in Figure 1, which includes the three sequential components: 1) *Graph Embedding Layer of Composite Code Semantics*, which encodes the raw source code of a function into a joint graph structure with comprehensive program semantics; 2) *Gated Graph Recurrent Layers*, which learn the features of nodes through aggregating and passing information on neighboring nodes in graphs; and 3) *the Conv module* that extracts meaningful node representation for graph-level prediction.

### 2.1 Problem Formulation

Most machine learning or pattern based approaches predict vulnerability at the coarse granularity level of a source file or an application, i.e., whether a source file or an application is potentially vulnerable [10, 17, 13, 15]. Here we analyze vulnerable code at the *function level* which is a finer level of granularity in the overall flow of vulnerability analysis. We formalize the identification of vulnerable functions as a binary classification problem, i.e., learning to decide whether a given

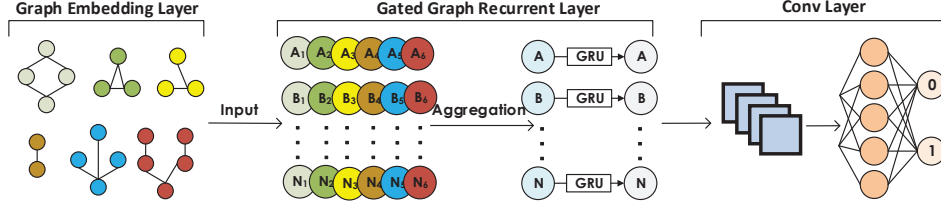

Figure 1: The Architecture of *Devign*

function in raw source code is vulnerable or not. Let a sample of data be defined as $((c_i, y_i)|c_i \in \mathcal{C}, y_i \in \mathcal{Y}), i \in \{1, 2, \ldots, n\}$, where $\mathcal{C}$ denotes the set of functions in code, $\mathcal{Y} = \{0, 1\}^n$ represents the label set with 1 for vulnerable and 0 otherwise, and $n$ is the number of instances. Since $c_i$ is a function, we assume it is encoded as a multi-edged graph $g_i(V, X, A) \in \mathcal{G}$ (See Section 2.2 for the embedding details). Let $m$ be the total number of nodes in $V$, $X \in \mathbb{R}^{m \times d}$ is the initial node feature matrix where each vertex $v_j$ in $V$ is represented by a $d$-dimensional real-valued vector $x_j \in \mathbb{R}^d$. $A \in \{0, 1\}^{k \times m \times m}$ is the adjacency matrix, where $k$ is the total number of edge types. An element $e_{s,t}^p \in A$ equal to 1 indicates that node $v_s, v_t$ is connected via an edge of type $p$, and 0 otherwise. The goal of *Devign* is to learn a mapping from $\mathcal{G}$ to $\mathcal{Y}$, $f : \mathcal{G} \mapsto \mathcal{Y}$ to predict whether a function is vulnerable or not. The prediction function $f$ can be learned by minimizing the loss function below:

$$\min \sum_{i=1}^{n} \mathcal{L}(f(g_i(V, X, A), y_i|c_i)) + \lambda \omega(f) \tag{1}$$

where $\mathcal{L}(\cdot)$ is the cross entropy loss function, $\omega(\cdot)$ is a regularization, and $\lambda$ is an adjustable weight.

## 2.2 Graph Embedding Layer of Composite Code Semantics

As illustrated in Figure 1, the graph embedding layer $EMB$ is a mapping from the function code $c_i$ to graph data structures as the input of the model, i.e.,

$$g_i(V, X, A) = EMB(c_i), \forall i = \{1, \ldots, n\} \tag{2}$$

In this section, we describe the motivation and method on why and how to utilize the classical code representations to embed the code into a composite graph for feature learning.

### 2.2.1 Classical Code Graph Representation and Vulnerability Identification

In program analysis, various representations of the program are utilized to manifest deeper semantics behind the textual code, where classic concepts include ASTs, control flow, and data flow graphs that capture the syntactic and semantic relationships among the different tokens of the source code. Majority of vulnerabilities such as memory leak are too subtle to be spotted without a joint consideration of the composite code semantics [17]. For example, it is reported that ASTs alone can be used to find only insecure arguments [17]. By combining ASTs with control flow graphs, it enables to cover two more types of vulnerabilities, i.e., resource leaks and some use-after-free vulnerabilities. By further integrating the three code graphs, it is possible to describe most types except two that need extra external information (i.e., race condition that depends on runtime properties and design errors that are hard to model without details on the intended design of a program)

Though the vulnerability templates in [17] are *manually* crafted in the form of graph traversals, it conveyed the key insight and proved the feasibility to learn a broader range of vulnerability patterns through integrating properties of ASTs, control flow graphs and data flow graphs into a joint data structure. Besides the three classical code structures, we also take the natural sequence of source code into consideration, since the recent advance on deep learning based vulnerability detection has demonstrated its effectiveness [13, 14]. It can complement the classical representations because its unique flat structure captures the relationships of code tokens in a 'human-readable' fashion.

### 2.2.2 Graph Embedding of Code

Next we briefly introduce each type of the code representations and how we represent various subgraphs into one joint graph, following a code example of integer overflow as in Figure 2(a) and its graph representation as shown in Figure 2(b).

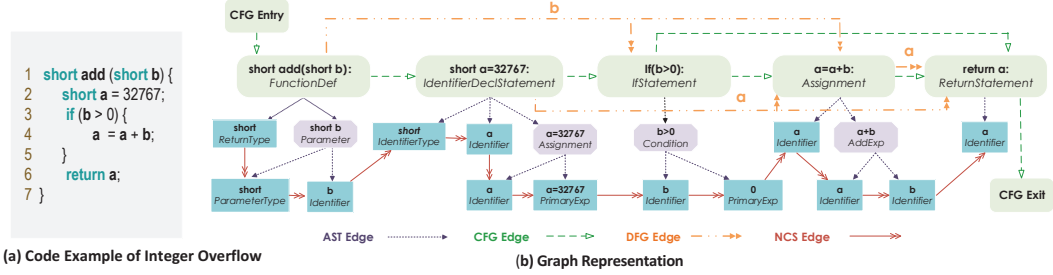

(a) Code Example of Integer Overflow    (b) Graph Representation

Figure 2: Graph Representation of Code Snippet with Integer Overflow

**Abstract Syntax Tree (AST)** AST is an ordered tree representation structure of source code. Usually, it is the first-step representation used by code parsers to understand the fundamental structure of the program and to examine syntactic errors. Hence, it forms the basis for the generation of many other code representations and the node set of AST $V^{ast}$ includes all the nodes of the rest three code representations used in this paper. Starting from the root node, the codes are broken down into code blocks, statements, declaration, expressions and so on, and finally into the primary tokens that form the leaf nodes. The major AST nodes are shown in Figure 2. All the boxes are AST nodes, with specific codes in the first line and node type annotated. The blue boxes are leaf nodes of AST and purple arrows represent the child-parent *AST* relations.

**Control Flow Graph (CFG)** CFG describes all paths that might be traversed through a program during its execution. The path alternatives are determined by conditional statements, e.g., *if*, *for*, and *switch* statements. In CFGs, nodes denote statements and conditions, and they are connected by directed edges to indicate the transfer of control. The *CFG* edges are highlighted with green dashed arrows in Figure 2. Particularly, the flow starts from the entry and ends at the exit, and two different paths derive at the *if* statements.

**Data Flow Graph (DFG)** DFG tracks the usage of variables throughout the CFG. Data flow is variable oriented and any data flow involves the access or modification of certain variables. A DFG edge represents the subsequent access or modification onto the same variables. It is illustrated by orange double arrows in Figure 2 and with the involved variables annotated over the edge. For example, the parameter $b$ is used in both the *if* condition and the assignment statement.

**Natural Code Sequence (NCS)** In order to encode the natural sequential order of the source code, we use *NCS* edges to connect neighboring code tokens in the ASTs. The main benefit with such encoding is to reserve the programming logic reflected by the sequence of source code. The *NCS* edges are denoted by red arrows in Figure 2, connect all the leaf nodes of the AST.

Consequently, a function $c_i$ can be denoted by a joint graph $g$ with the four types of subgraphs (or $4$ types of edges) sharing the same set of nodes $V = V^{ast}$. As shown in Figure (2), every node $v \in V$ has two attributes, *Code* and *Type*. *Code* contains the source code represented by $v$, and the type of $v$ denotes the type attribute. The initial node representation $x_v$ shall reflect the two attributes. Hence, we encode *Code* by using a pre-trained word2vec model with the code corpus built on the whole source code files in the projects, and *Type* by label encoding. We concatenate the two encodings together as the initial node representation $x_v$.

### 2.3 Gated Graph Recurrent Layers

The key idea of graph neural networks is to embed node representation from local neighborhoods through the neighborhood aggregation. Based on the different techniques for aggregating neighborhood information, there are graph convolutional networks [19], GraphSAGE [20], gated graph recurrent networks [18] and their variants. We chose the gated graph recurrent network to learn the node embedding, because it allows to go deeper than the other two and is more suitable for our data with both semantics and graph structures [21].

Given an embedded graph $g_i(V, X, A)$, for each node $v_j \in V$, we initialize the node state vector $h_j^{(1)} \in \mathbb{R}^z, z \geq d$ using the initial annotation by copying $x_j$ into the first dimensions and padding extra 0's to allow hidden states that are larger than the annotation size, i.e., $h_j^1 = [x_j^\top, \mathbf{0}]^\top$. Let $T$ be the total number of time-step for neighborhood aggregation. To propagate information throughout graphs, at each time step $t \leq T$, all nodes communicate with each other by passing information via

edges dependent on the edge type and direction (described by the $p^{th}$ adjacent matrix $A_p$ of $A$, from the definition we can find that the number of adjacent matrix equals to edge types), i.e.,

$$a_{j,p}^{(t-1)} = A_p^\top \left( W_p \left[ h_1^{(t-1)\top}, \ldots, h_m^{(t-1)\top} \right] + b \right) \tag{3}$$

where $W_p \in \mathbb{R}^{z \times z}$ is the weight to learn and $b$ is the bias. In particular, a new state $a_{j,p}$ of node $v_j$ is calculated by aggregating information of all neighboring nodes defined on the adjacent matrix $A_p$ on edge type $p$. The remaining steps are gated recurrent unit (GRU) that incorporate information from all types with node $v$ and the previous time step to get the current node's hidden state $h_{i,v}^{(t)}$, i.e.,

$$h_j^{(t)} = GRU(h_j^{(t-1)}, AGG(\{a_{j,p}^{(t-1)}\}_{p=1}^k)) \tag{4}$$

where $AGG(\cdot)$ denotes an aggregation function that could be one of the functions $\{MEAN, MAX, SUM, CONCAT\}$ to aggregate the information from different edge types to compute the next time-step node embedding $h^{(t)}$. We use the $SUM$ function in the implementation. The above propagation procedure iterates over $T$ time steps, and the state vectors at the last time step $H_i^{(T)} = \{h_j^{(T)}\}_{j=1}^m$ is the final node representation matrix for the node set $V$.

## 2.4 The Conv Layer

The generated node features from the gated graph recurrent layers can be used as input to any prediction layer, e.g., for node or link or graph-level prediction, and then the whole model can be trained in an end-to-end fashion. In our problem, we require to perform the task of graph-level classification to determine whether a function $c_i$ is vulnerable or not. The standard approach to graph classification is gathering all these generated node embeddings globally, e.g., using a linear weighted summation to flatly adding up all the embeddings [18, 22] as shown in Eq (5),

$$\tilde{y}_i = Sigmoid\left( \sum MLP([H_i^{(T)}, x_i]) \right) \tag{5}$$

where the $sigmoid$ function is used for classification and $MLP$ denotes a Multilayer Perceptron (MLP) that maps the concatenation of $H_i^{(T)}$ and $x_i$ to a $\mathbb{R}^m$ vector. This kind of approach hinders effective classification over entire graphs [23, 24].

Thus, we design the *Conv* module to select sets of nodes and features that are relevant to the current graph-level task. Previous works in [24] proposed to use a SortPooling layer after the graph convolution layers to sort the node features in a consistent node order for graphs without fixed ordering, so that traditional neural networks can be added after it and trained to extract useful features characterizing the rich information encoded in graph. In our problem, each code representation graph has its own predefined order and connection of nodes encoded in the adjacent matrix, and the node features are learned through gated recurrent graph layers instead of graph convolution networks that requires to sort the node features from different channels. Therefore, we directly apply 1-D convolution and dense neural networks to learn features relevant to the graph-level task for more effective prediction[1]. We define $\sigma(\cdot)$ as a 1-D convolutional layer with maxpooling, then

$$\sigma(\cdot) = MAXPOOL\big(Relu\big(CONV(\cdot)\big)\big) \tag{6}$$

Let $l$ be the number of convolutional layers applied, then the *Conv* module, can be expressed as

$$Z_i^{(1)} = \sigma\big([H_i^{(T)}, x_i]\big), \ldots, Z_i^{(l)} = \sigma\big(Z_i^{(l-1)}\big) \tag{7}$$

$$Y_i^{(1)} = \sigma\big(H_i^{(T)}\big), \ldots, Y_i^{(l)} = \sigma\big(Y_i^{(l-1)}\big) \tag{8}$$

$$\tilde{y}_i = Sigmoid\big(AVG(MLP(Z_i^{(l)}) \odot MLP(Y_i^{(l)}))\big) \tag{9}$$

where we firstly apply traditional 1-D convolutional and dense layers respectively on the concatenation $[H_i^{(T)}, x_i]$ and the final node features $H_i^{(T)}$, followed by a pairwise multiplication on the two outputs, then an average aggregation on the resulted vector, and at last make a prediction.

## 3 Evaluation

We evaluate the benefits of *Devign* against a number of state-of-the-art vulnerability discovery methods, with the goal of understanding the following questions:

Table 1: Data Sets Overview

| Project | Sec. Rel. Commits | VFCs | Non-VFCs | Graphs | Vul Graphs | Non-Vul Graphs |
|---|---|---|---|---|---|---|
| Linux Kernel | 12811 | 8647 | 4164 | 16583 | 11198 | 5385 |
| QEMU | 11910 | 4932 | 6978 | 15645 | 6648 | 8997 |
| Wireshark | 10004 | 3814 | 6190 | 20021 | 6386 | 13635 |
| FFmpeg | 13962 | 5962 | 8000 | 6716 | 3420 | 3296 |
| Total | 48687 | 23355 | 25332 | 58965 | 27652 | 31313 |

**Q1** How does our *Devign* compare to the other learning based vulnerability identification methods?

**Q2** How does our *Conv* module powered *Devign* compare to the *Ggrn* with the flat summation in Eq (5) for the graph-level classification task?

**Q3** Can *Devign* learn from each type of the code representations (e.g., a single-edged graph with one type of information)? And how do the *Devign* models with the composite graphs (e.g., all types of code representations) compare to each of the single-edged graphs?

**Q4** Can *Devign* have a better performance compared to some static analyzers in the real scenario where the dataset is imbalanced with an extremely low percentage of vulnerable functions?

**Q5** How does *Devign* perform on the latest vulnerabilities reported publicly through CVEs?

## 3.1 Data Preparation

It is never trivial to obtain high-quality data sets of vulnerable functions due to the demand of qualified expertise. We noticed that despite [15] released data sets of vulnerable functions, the labels are generated by statistic analyzers which are not accurate. Other potential datasets used in [25] are not available. In this work, supported by our industrial partners, we invested a team of security to collect and label the data from scratch. Besides raw function collection, we need to generate graph representations for each function and initial representations for each node in a graph. We describe the detailed procedures below.

**Raw Data Gathering** To test the capability of *Devign* in learning vulnerability patterns, we evaluate on manually-labeled functions collected from 4 large C-language open-source projects that are popular among developers and diversified in functionality, i.e., Linux Kernel, QEMU, Wireshark, and FFmpeg.

To facilitate and ensure the quality of data labelling, we started by collecting security-related commits which we would label as vulnerability-fix commits or non-vulnerability fix commits, and then extracted vulnerable or non-vulnerable functions directly from the labeled commits. The vulnerability-fix commits (VFCs) are commits that fix potential vulnerabilities, from which we can extract vulnerable functions from the source code of versions previous to the revision made in the commits. The non-vulnerability-fix commits (non-VFCs) are commits that do not fix any vulnerability, similarly from which we can extract non-vulnerable functions from the source code before the modification. We adopted the approach proposed in [26] to collect the commits. It consists of the following two steps. 1) *Commits Filtering*. Since only a tiny part of commits are vulnerability related, we exclude the security-unrelated commits whose messages are not matched by a set of security-related keywords such as DoS and injection. The rest, more likely security-related, are left for manual labelling. 2) *Manual Labelling*. A team of four professional security researchers spent totally *600 man-hours* to perform a two round data labelling and cross-verification.

Given a VFC or non-CFC, based on the modified functions, we extract the source code of these functions before the commit is applied, and assign the labels accordingly.

**Graph Generation** We make use of the open-source code analysis platform for C/C++ based on code property graphs, Joern [17], to extract ASTs and CFGs for all functions in our data sets. Due to some inner compile errors and exceptions in Joern, we can only obtain ASTs and CFGs for part of functions. We filter out these functions without ASTs and CFGs or with oblivious errors in ASTs and CFGs. Since the original DFGs edges are labeled with the variables involved, which tremendously increases the number of the types of edges and meanwhile complicates embedded graphs, we substitute the DFGs with three other relations, *LastRead (DFG_R)*, *LastWrite (DFG_W)*, and *ComputedFrom (DFG_C)* [27], to make it more adaptive for the graph embedding. *DFG_R* represents the immediate last read of each occurrence of the variable. Each occurrence can be directly recognized from the leaf nodes of ASTs. *DFG_W* represents the immediate last write of each occurrence of variables. Similarly, we make these annotations to the leaf node variables. *DFG_C* determines the sources of a

Table 2: Classification accuracies and F1 scores in percentages: The two far-right columns give the maximum and average relative difference in accuracy/F1 compared to *Devign* model with the composite code representations, i.e., *Devign* (Composite).

| Method | Linux Kernel ACC | F1 | QEMU ACC | F1 | Wireshark ACC | F1 | FFmpeg ACC | F1 | Combined ACC | F1 | Max Diff ACC | F1 | Avg Diff ACC | F1 |
|---|---|---|---|---|---|---|---|---|---|---|---|---|---|---|
| Metrics + Xgboost | 67.17 | 79.14 | 59.49 | 61.27 | 70.39 | 61.31 | 67.17 | 63.76 | 61.36 | 63.76 | 14.84 | 11.80 | 10.30 | 8.71 |
| 3-layer BiLSTM | 67.25 | 80.41 | 57.85 | 57.75 | 69.08 | 55.61 | 53.27 | 69.51 | 59.40 | 65.62 | 16.48 | 15.32 | 14.04 | 8.78 |
| 3-layer BiLSTM + Att | 75.63 | 82.66 | 65.79 | 59.92 | 74.50 | 58.52 | 61.71 | 66.01 | 69.57 | 68.65 | 8.54 | 13.15 | 5.97 | 7.41 |
| CNN | 70.72 | 79.55 | 60.47 | 59.29 | 70.48 | 58.15 | 53.42 | 66.58 | 63.36 | 60.13 | 16.16 | 13.78 | 11.72 | 9.82 |
| *Ggrn* (AST) | 72.65 | 81.28 | 70.08 | 66.84 | 79.62 | 64.56 | 63.54 | 70.43 | 67.74 | 64.67 | 6.93 | 8.59 | 4.69 | 5.01 |
| *Ggrn* (CFG) | 78.79 | 82.35 | 71.42 | 67.74 | 79.36 | 65.40 | 65.00 | 71.79 | 70.62 | 70.86 | 4.58 | 5.33 | 2.38 | 2.93 |
| *Ggrn* (NCS) | 78.68 | 81.84 | 72.99 | 69.98 | 78.13 | 59.80 | 65.63 | 69.09 | 70.43 | 69.86 | 3.95 | 8.16 | 2.24 | 4.45 |
| *Ggrn* (DFG_C) | 70.53 | 81.03 | 69.30 | 56.06 | 73.17 | 50.83 | 63.75 | 69.44 | 65.52 | 64.57 | 9.05 | 17.13 | 6.96 | 10.18 |
| *Ggrn* (DFG_R) | 72.43 | 80.39 | 68.63 | 56.35 | 74.15 | 52.25 | 63.75 | 71.49 | 66.74 | 62.91 | 7.17 | 16.72 | 6.27 | 9.88 |
| *Ggrn* (DFG_W) | 71.09 | 81.27 | 71.65 | 65.88 | 72.72 | 51.04 | 64.37 | 70.52 | 63.26 |  | 9.21 | 16.92 | 6.84 | 8.17 |
| *Ggrn* (Composite) | 74.55 | 79.93 | 72.77 | 66.25 | 78.79 | 67.32 | 64.46 | 70.33 | 70.35 | 69.37 | 5.12 | 6.82 | 3.23 | 3.92 |
| *Devign* (AST) | **80.24** | 84.57 | 71.31 | 65.19 | 79.04 | 64.37 | 65.63 | 71.83 | 69.21 | 69.99 | 3.95 | 7.88 | 2.33 | 3.37 |
| *Devign* (CFG) | 80.03 | 82.91 | 74.22 | 70.73 | 79.62 | 66.05 | 66.89 | 70.22 | 71.32 | 71.27 | 2.69 | 3.33 | 1.00 | 2.33 |
| *Devign* (NCS) | 79.58 | 81.41 | 72.32 | 68.98 | 79.75 | 65.88 | 67.29 | 68.89 | 70.82 | 68.45 | 2.29 | 4.81 | 1.46 | 3.84 |
| *Devign* (DFG_C) | 78.81 | 83.87 | 72.30 | 70.62 | 79.95 | 66.47 | 65.83 | 70.12 | 69.88 | 70.21 | 3.75 | 3.43 | 2.06 | 2.30 |
| *Devign* (DFG_R) | 78.25 | 80.33 | 73.77 | 70.60 | 80.66 | 66.17 | 66.46 | 72.12 | 71.49 | 71.49 | 3.12 | 4.64 | 1.29 | 2.53 |
| *Devign* (DFG_W) | 78.70 | 84.21 | 72.54 | 71.08 | 80.59 | 66.68 | 67.50 | 70.86 | 71.41 | 71.14 | 2.08 | 2.69 | 1.27 | 1.77 |
| *Devign* (Composite) | 79.58 | **84.97** | **74.33** | **73.07** | **81.32** | **67.96** | **69.58** | **73.55** | **72.26** | **73.26** | - | - | - | - |

Table 3: Classification accuracies and F1 scores in percentages under the real imbalanced setting

| Method | Cppcheck ACC | F1 | Flawfinder ACC | F1 | CXXX ACC | F1 | 3-layer BiLSTM ACC | F1 | 3-layer BiLSTM + Att ACC | F1 | CNN ACC | F1 | *Devign* (Composite) ACC | F1 |
|---|---|---|---|---|---|---|---|---|---|---|---|---|---|---|
| Linux | 75.11 | 0 | 78.46 | 12.57 | 19.44 | 5.07 | 18.25 | 13.12 | 8.79 | 16.16 | 29.03 | 15.38 | 69.41 | **24.64** |
| QEMU | 89.21 | 0 | 86.24 | 7.61 | 33.64 | 9.29 | 29.07 | 15.54 | 78.43 | 10.50 | 75.88 | 18.80 | 89.27 | **41.12** |
| Wireshark | 89.19 | 10.17 | 89.92 | 9.46 | 33.26 | 3.95 | 91.39 | 10.75 | 84.90 | 28.35 | 86.09 | 8.69 | 89.37 | **42.05** |
| FFmpeg | 87.72 | 0 | 80.34 | 12.86 | 36.04 | 2.45 | 11.17 | 18.71 | 8.98 | 16.48 | 70.07 | 31.25 | 69.06 | **34.92** |
| Combined | 85.41 | 2.27 | 85.65 | 10.41 | 29.57 | 4.01 | 9.65 | 16.59 | 15.58 | 16.24 | 72.47 | 17.94 | 75.56 | **27.25** |

variable. In an assignment statement, the left-hand-side (lhs) variable is assigned with a new value by the right-hand-side (rhs) expression. DFG_C captures such relations between the lhs variable and each of the rhs variable. Further, we remove functions with node size greater than 500 for computational efficiency, which accounts for 15%. We summarize the statistics of the data sets in Table 1.

## 3.2 Baseline Methods

In the performance comparison, we compare *Devign* with the state-of-the-art machine-learning-based vulnerability prediction methods, as well as the gated graph recurrent network (*Ggrn*) that used the linearly weighted summation for classification.

**Metrics + Xgboost** [25]: We collect totally 4 complexity metrics and 11 vulnerability metrics for each function using Joern, and utilize Xgboost for classification. Here we did not use the proposed binning and ranking method because it was not learning based, but a heuristic designed to rank the likelihood of being vulnerable for the full functions in a project. We search the best parameters via Bayes Optimization [28].

**3-layer BiLSTM** [13]: It treats the source code as natural languages and input the tokenized code into bidirectional LSTMs with initial embeddings trained via Word2vec. Here we implemented a 3-layer bidirectional for the best performance.

**3-layer BiLSTM + Att:** It is an improved version of [13] with the attention mechanism [29].

**CNN** [14]: Similar to [13], it takes source code as natural languages and utilizes the bag of words to get the initial embeddings of code tokens, and then feeds them to CNNs to learn.

## 3.3 Performance Evaluation

*Devign* **Configuration** In the embedding layer, the dimension of word2vec for the initial node representation is 100. In the gated graph recurrent layer, we set the the dimension of hidden states as 200, and number of time steps as 6. For the *Conv* parameters of *Devign*, we apply (1, 3) filter with

ReLU activation function for the first convolution layer which is followed by a max pooling layer with (1, 3) filter and (1, 2) stride, and (1, 1) filter for the second convolution layer with a max pooling layer with (2, 2) filter and (1, 2) stride. We use the Adam optimizer with learning rate 0.0001 and batch size 128, and $L2$ regularization to avoid overfitting. We randomly shuffle each dataset and split 75% for the training and the rest 25% for validation. We train our model on Nvidia Graphics Tesla M40 and P40, with 100-epoch patience for early stopping.

**Results Analysis** We use *accuracy* and *F1 score* to measure performance. Table 2 summarizes all the experiment results. First, we analyze the results regarding **Q1**, the performance of *Devign* with other learning based methods. From the results on baseline methods, *Ggrn* and *Devign* with composite code representations, we can see that both *Ggrn* and *Devign* significantly outperform the baseline methods in all the data sets. Especially, compared to all the baseline methods, the relative accuracy gain by *Devign* is averagely 10.51%, at least 8.54% on the QEMU dataset. *Devign* (Composite) outperforms the 4 baseline methods in terms of F1 score as well, i.e., the relative gain of F1 score is 8.68% on the average and the minimum relative gains on each dataset (Linux Kernel, QEMU, Wirshark, FFmpeg and Combined) are 2.31%, 11.80%, 6.65%, 4.04% and 4.61% respectively. As Linux follows best practices of coding style, the F1 score 84.97 by *Devign* is the highest among all datasets. *Hence,* Devign *with comprehensive semantics encoded in graphs performs significantly better than the state-of-the-art vulnerability identification methods.*

Next, we investigate the answer to **Q2** about the performance gain of *Devign* against *Ggrn*. We first look at the score with the composite code representation. It demonstrates that, in all the data sets, *Devign* reaches higher accuracy (an average of 3.23%) than *Ggrn*, where the highest accuracy gain is 5.12% on the FFmpeg data set. Also *Devign* gets better F1, an average of 3.92% higher than *Ggrn*, where the highest F1 gain is 6.82 % on the QEMU data set. Meanwhile, we look at the score with each single code representation, from which, we get similar conclusion that generally *Devign* significantly outperforms *Ggrn*, where the maximum accuracy gain is 9.21% for the DFG_W edge and the maximum F1 gain is 17.13% for the DFG_C. *Overall the average accuracy and F1 gain by* Devign *compared with* Ggrn *are 4.66%, 6.37% among all cases, which indicates the Conv module extracts more related nodes and features for graph-level prediction.*

Then we check the results for **Q3** to answer whether *Devign* can learn different types of code representation and the performance on composite graphs. Surprisingly we find that the results learned from single-edged graphs are quite encouraging in both of *Ggrn* and *Devign*. For *Ggrn*, we find that the accuracy in some specific types of edges is even slightly higher than that in the composite graph, e.g., both CFG and NCS graphs have better results on the FFmpeg and combined data set. For *Devign*, in terms of accuracy, except the Linux data set, the composite graph representation is overall superior to any single-edged graph with the gain ranging from 0.11% to 3.75%. In terms of F1 score, the improvement brought by composite graph compared with the single-edged graphs is averagely 2.69%, ranging from 0.4% to 7.88% in the *Devign* in all tests. *In summary, composite graphs help* Devign *to learn better prediction models than single-edged graphs.*

To answer **Q4** about the comparison with static analyzers on the real imbalanced dataset, we randomly sampled the test data to create imbalanced datasets with 10% vulnerable functions according to a large industrial analysis [26]. We compare with the well-known open-source static analyzers Cppcheck, Flawfinder, and a commercial tool CXXX which we hide the name for legal concern. The results are shown in Table 3, where our approach outperforms significantly with a 27.99% higher average F1 score compared with the performance of all the analyzers and on all the datasets (individual and combined). Meanwhile, static analyzers tend to miss most vulnerable functions and have high false positives, e.g., Cppcheck found 0 vulnerability in 3 out of the 4 single project datasets.

Finally to answer **Q5** on the latest exposed vulnerabilities, we scrape the latest 10 CVEs of each project respectively to check whether *Devign* can be potentially applied to identify zero-day vulnerabilities. Based on commit fix of the 40 CVEs, we totally get 112 vulnerable functions. *We input these functions into the trained* Devign *model and achieve an average accuracy of 74.11%, which manifests* Devign*'s potentiality of discovering new vulnerabilities in practical applications.*

## 4   Related Work

The success of deep learning has inspired the researchers to apply it for more automated solutions to vulnerability discovery on source code [15, 13, 14]. The recent works [13, 15, 14] treat source code

as flat natural language sequences, and explore the potential of natural language process techniques in vulnerability detection. For instance, [15, 13] built models upon LSTM/BiLSTM neural networks, while [14] proposed to use the CNNs instead.

To overcome the limitations of the aforementioned models on expressing logic and structures in code, a number of works have attempted to probe more structural neural networks such as tree structures [30] or graph structures [18, 31, 27] for various tasks. For instance, [18] proposed to generate logical formulas for program verification through gated graph recurrent networks, and [27] aimed at prediction of variable names and variable miss-usage. [31] proposed Gemini for binary code similarity detection, where functions in binary code are represented by attributed control flow graphs and input Structure2vec [22] for learning graph embedding. Different from all these works, our work targeted at vulnerability identification, and incorporated comprehensive code representations to express as many types of vulnerabilities as possible. Beside, our work adopt gated graph recurrent layers in [18] to consider semantics of nodes (e.g., node annotations) as well as the structural features, both of which are important in vulnerability identification. Structure2vec focuses primarily on learning structural features. Compared with [27] that applies gated graph recurrent network for variable prediction, we explicitly incorporate control flow graph into the composite graph and propose the *Conv* module for efficient graph-level classification.

## 5    Conclusion and Future Work

We introduce a novel vulnerability identification model *Devign* that is able to encode a source-code function into a joint graph structure from multiple syntax and semantic representations and then leverage the composite graph representation to effectively learn to discover vulnerable code. It achieved a new state of the art on machine-learning-based vulnerable function discovery on real open-source projects. Interesting future works include efficient learning from big functions via integrating program slicing, applying the learnt model to detect vulnerabilities cross projects, and generating human-readable or explainable vulnerability assessment.

## Acknowledgements

This work was supported by Alibaba-NTU JRI project (M4062640.J4A), Security: A Compositional Approach Of Building Security Verified System (M4192001.023.710079) and National Research Foundation, Prime Ministers Office, Singapore Under its National Cybersecurity R&D Program (Award No. NRF2018NCR-NCR005-001).

## Footnotes

[1]We also tried LSTMs and BiLSTMs (with and without attention mechanisms) on the sorted nodes in AST order, however, the convolution networks work best overall.

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
