[Reviews · NeurIPS 2019]

Reviewer 1



Originality: the proposed method in this paper is original. It focuses on vulnerability identification in software systems. Quality: this work is almost technological sound. Honestly, I have no research experience on vulnerability identification but the problem is well-defined and is modeled as a machine learning problem in this paper. The whole process is clear and the motivation is clearly described. I prefer to see this work in the application section of NeurIPS. Clarity: This paper is well organized. Significance: It is clear that other researchers can use the ideas of Devign, especially the graph embedding approach of this paper. It considers the semantics of nodes and structural features in vulnerability identification.

Reviewer 2



This paper addresses the prediction of functions with vulnerabilities. The dataset used is based on real-world applications (e.g., Linux (kernel?), QEMU). The method employed represents code by combining different static code analysis techniques. All static graphs are well-known, however I haven't seen all of them combined in one graph for representing code. The graphs used are ASTs, CFG, DFG (with some simplifications, please see comments in improvements), in addition to the code treated as a sequence of tokens. The paper is well written, generally straightforward to read and has sufficient background information on graph embeddings of code and the various graphs employed such that a reader outside of the domain can follow the discussion. The results obtained are quite promising. In particular, I appreciated how the authors look at the latest commits in the projects they used in the dataset to understand whether their trained model could deal with code "in real time".

Reviewer 3



The main contribution of this paper is a manually curated dataset of functions determining if a function is vulnerable or benign. The novelty here is that there is no bias introduced by either assuming that most of the data is correct (assumed by anomaly detection works like e.g. [19]) or encoding the bias of an existing static analyzer. The evaluation results on this datasets, however, are not convincing for practical application of the resulting classifier. The training data has similar number of vulnerable and benign graphs, while practical programs have much lower percentage of vulnerable functions than the accuracy of the classifier. Thus, accuracy in the 70-80% range is not practical and likely its output in practice will look like pure noise (if 2 out of 100 functions are vulnerable, a classifier with 70% accuracy will give on average 28-29 false positives and has non-trivial chance to miss a vulnerability). This means that the classifier needs significant changes. Furthermore, the baseline to which the paper should compare is static analysis, not other neural architectures. Technically the paper is almost a verbatim copy of the architecture already proposed in [19] for prediction of (variable) names in programs. The edge types described in Figure 2 are also quite similar to the ones in [19]. Unfortunately, the writing of the paper makes it appear as if these architectural details are contributions of this work. The evaluation also shows that the neural network captures program-specific features that do not transfer between programs. The combined accuracy of the classifiers is lower than most of the individual classifiers. This means that collecting more training data will not help, but may actually harm the classifiers. It also jeopardizes the paper claims of capturing semantic features. minor: pg.6 s/non-CFC/non-VFC/ - UPDATE - The authors have mostly answered my concerns with the extra experiments. The accuracy metric gives little information about the recall or the precision of the learned code analyzer. It would help to provide precision instead of accuracy in the evaluation. In terms of differences from [19], partial control flow is also encoded there. Also other papers encode control flow. It would help if the paper focuses more on empirical evaluation (as with the provided extra experiments) and introducing the new task, because the actual graph architecture is hard to differentiate from prior works.

[Author Response · NeurIPS 2019]

Table 1: Classification accuracies and F1 scores in percentiles under the imbalanced setting

| Method | Cppcheck ACC | F1 | Flawfinder ACC | F1 | CXXX ACC | F1 | 3-layer BiLSTM ACC | F1 | 3-layer BiLSTM + Att ACC | F1 | CNN ACC | F1 | *Devign* (Composite) ACC | F1 |
|---|---|---|---|---|---|---|---|---|---|---|---|---|---|---|
| Linux | 75.11 | 0 | 78.46 | 12.57 | 19.44 | 5.07 | 18.25 | 13.12 | 8.79 | 16.16 | 29.03 | 15.38 | 69.41 | **24.64** |
| QEMU | 89.21 | 0 | 86.24 | 7.61 | 33.64 | 9.29 | 29.07 | 15.54 | 78.43 | 10.50 | 75.88 | 18.80 | 89.27 | **41.12** |
| Wireshark | 89.19 | 10.17 | 89.92 | 9.46 | 33.26 | 3.95 | 91.39 | 10.75 | 84.90 | 28.35 | 86.09 | 8.69 | 89.37 | **42.05** |
| FFmpeg | 87.72 | 0 | 80.34 | 12.86 | 36.04 | 2.45 | 11.17 | 18.71 | 8.98 | 16.48 | 70.07 | 31.25 | 69.06 | **34.92** |
| Combined | 85.41 | 2.27 | 85.65 | 10.41 | 29.57 | 4.01 | 9.65 | 16.59 | 15.58 | 16.24 | 72.47 | 17.94 | 75.56 | **27.25** |

Table 2: Classification accuracies and F1 scores of *Devign* as data size increases

| Data size | Linux ACC | F1 | QEMU ACC | F1 | Wireshark ACC | F1 | FFmpeg ACC | F1 | Combined ACC | F1 |
|---|---|---|---|---|---|---|---|---|---|---|
| 1/3 | 67.29 | 76.58 | 66.96 | 64.26 | 71.37 | 52.65 | 61.88 | 67.67 | 66.64 | 67.80 |
| 2/3 | 74.67 | 83.51 | 72.10 | 66.17 | 75.58 | 54.32 | 65.58 | 71.72 | 68.42 | 68.52 |
| whole dataset | 79.58 | 84.97 | 74.33 | 73.07 | 81.32 | 67.96 | 69.58 | 73.55 | 72.26 | 73.26 |

**Reviewer 1 :**  Thanks for the valuable comments. This work aims at encoding rich semantic information of program into the neural networks. The semantics of programs are typically captured via AST, CFG, DFG, which are in graph structure. GNN can naturally encode these semantic information. We will include more description about why we chose graph embedding and the motivation of our approach in the next revision.

**Reviewer 2:**  Thanks for the valuable comments. 1) The "Linux" in our dataset means "Linux Kernel". 2) We will include the line numbers in graph generation subsection, repeat the questions that the experiments address, add a table for Q4, and summarize the hyper-parameters in a separate subsection of "Devign Configuration" in the future revision. 3) We have made our dataset public available in our website.

**Reviewer 3:**  Thanks for the valuable comments and questions. 1) We understand the reviewer's concern that the ratio of vulnerable and non-vulnerable functions in our dataset is relatively balanced compared to practical applications. Directly using our model, as well as any other trained model to classify more imbalanced data may affect the performance. A practically usable trained model has to be customized and tuned specifically to each application and data set case by case. Besides, there are various methods specially for data imbalance to alleviate the issues. Due to time limit, we cannot incorporate these techniques and retrain models, but we conducted experiments on using our trained models to predict under the imbalanced setting. A large industrial scale analysis in [1] shows that vulnerable functions is around 10% of total functions, therefore we randomly sampled the test data to create imbalanced datasets with 10% vulnerable functions. The results are shown in Table 1, where our approach achieves much better performance with an F1 score averagely **17.03** higher than all the machine learning methods under the same imbalanced data setting.

2) **Comparison with static analyzers**:  We compare with the well-known open-source static analyzers Cppcheck, Flawfinder and a commercial tool CXXX which we hide the name for legal concern. Table 1 shows the results, where our approach outperforms significantly all static analyzers with an F1 score **27.99** higher in the imbalanced setting. Static analyzers tend to miss most vulnerable functions and have high false positives, e.g., Cppcheck found 0 vulnerability in 3 out of the 4 single project datasets.

3) **Difference with [19]**: We focus on applying the GNN to learn the representation of vulnerable functions, which is same as [19] did to use it to learn for variable prediction. The AST, Control Flow and Data Flow edges we used are classical code property graph representations in programming analysis. We did not take this alone as a contribution, but did explore and find that applying all these edges help to learn better generally. One important note is that [19] didn't introduce the control flow graph (CFG), which is crucial in vulnerability analysis. We compared our method with the edges selected in [19] and found the performance without CFG is much worse than the one with CFG, i.e., accuracy 68.96 and F1 65.12 without CFG v.s. accuracy **72.26** and F1 **73.26** with CFG.

4) **Learning programming semantics across projects**: We don't think it can be simply concluded that no-semantic meaning can be learned across projects because the accuracy on the combined dataset is slightly lower than the average of the 4 datasets. The data from the 4 projects are too diversified from each other in terms of functionality, major vulnerability types, and root causes. Thus the diversity of the vulnerabilities in the combined datasets is much wider than each single data set. To deal with the diversity in the combined data, we believe that more data required to improve the overall performance. To verify it, we tested trained models with different sizes of the combined dataset, i.e., 1/3, 2/3 and all of the combined dataset. As shown in Table 2, both accuracy and F1 increases as the data volume increases. In addition to data diversity, the data size of each project also causes imbalance in the combined data set, which further impacts the overall performance.

[1] Automated identification of security issues from commit messages and bug reports. FSE 2017.

[Meta-Review · NeurIPS 2019]

The paper describes a technique for identifying functions in code that may suffer vulnerabilities. The problem is important and timely. The technique is evaluated on real data, and the results seem very promising. The reviewers liked the work and appreciated the response made by the authors.